# Intimal Sarcoma with MDM2/CDK4 Amplification and p16 Overexpression: A Review of Histological Features in Primary Tumor and Xenograft, with Immunophenotype and Molecular Profiling

**DOI:** 10.3390/ijms24087535

**Published:** 2023-04-19

**Authors:** Francisco Giner, Isidro Machado, Luis Alberto Rubio-Martínez, José Antonio López-Guerrero, Reyes Claramunt-Alonso, Samuel Navarro, Antonio Ferrández, Empar Mayordomo-Aranda, Antonio Llombart-Bosch

**Affiliations:** 1Pathology Department, University of Valencia, 46010 Valencia, Spain; 2Pathology Department, Hospital Universitari i Politècnic La Fe of Valencia, 46026 Valencia, Spain; 3Pathology Department, Instituto Valenciano de Oncología, 46009 Valencia, Spain; 4Molecular Biology Department, Instituto Valenciano de Oncología, 46009 Valencia, Spain; 5Pathology Department, Hospital Clínic Universitari, 46010 Valencia, Spain

**Keywords:** intimal sarcoma, immunohistochemistry, molecular alterations, therapeutic targets

## Abstract

Intimal sarcomas (IS) are rare malignant mesenchymal tumors arising in large blood vessels of the systemic and pulmonary circulation and also in the heart. They are morphologically similar to other spindle cell, poorly differentiated sarcomas. The prognosis is poor and depends mainly on surgical options. Three cases of IS were collected from two institutions. Clinical data were retrieved and histological study was performed. A wide immunohistochemical panel was analyzed. FISH of *MDM2* gene was performed, and a molecular study with NGS was implemented in all cases. The mean age of our cases was 54 years. Histologically, the tumors presented a diffuse growth pattern with heterogeneous atypical epithelioid or spindle cells and extensive thrombosed areas. All cases presented intense immunoexpression for MDM2, CDK4, CD117, c-myc, PDGFRA, and p16. PDGFRA, HTERT, and pan-TRK gained expression, while p16 lost intensity, being weaker in both the local recurrences and xenografts. The three cases showed amplification of *MDM2* by FISH. NGS analysis revealed amplifications in the *CDK4*, *PDGFRA*, and *KIT* genes, together with *BRAF* mutation and *KRAS* amplification. P16 was expressed in all cases, losing intensity in local recurrence and xenografts. Two new alterations, a *BRAF* mutation and a *KRAS* amplification, were detected by NGS in different tumors, opening up new therapeutic options for these patients.

## 1. Introduction

Intimal sarcomas (IS) are very rare malignant mesenchymal tumors arising in large blood vessels (Figure 1) of the systemic and pulmonary circulation and also in the heart [1]. The prognosis is poor and is influenced more by the surgical options than by the histologic subtype [2]. IS were first described by Mandelstamm in 1923 [3]. Since then, around 400 cases have been reported in the literature [4,5]. Few series have described these sarcomas [6], with the majority of publications referring to case reports. No xenografted tumors have been reported so far. The predominant clinical feature of IS is intraluminal growth, leading to local obstruction and seeding of emboli to other organs. In most cases, neoplasms grossly mimicking thrombi extend distally along the branches of involved vessels. IS are morphologically indistinguishable from other poorly differentiated sarcomas. Occasionally, myxoid areas or epithelioid features are apparent [7,8]. Sometimes IS may resemble leiomyosarcoma, hemangioendothelioma, synovial sarcoma, myxofibrosarcoma, or undifferentiated pleomorphic sarcoma. Other rare cases can display tumor cartilage or immature osteoid matrix; focal rhabdomyosarcomatous and angiosarcomatous features have also been described [6,7,9,10,11]. Immunohistochemically, variable positivity for smooth muscle actin and desmin has been found but never h-caldesmon. More recently, nuclear expression and amplification of MDM2 have been described in at least 70% of cases [12,13]. In other series, more than two-thirds were also positive for CDK4 and HMGA2, as well as myogenin, when a rhabdomyosarcomatous component was present. In addition, 14% of cases can be positive for cytokeratin, creating confusion with synovial sarcoma [6]. Genomic and molecular studies have shown amplification and activation of KIT and PDGFRA, suggesting therapeutic possibilities for tyrosine kinase receptor inhibitors [13]. Recent studies have communicated a genetic and epigenetic concordance of IS and undifferentiated pleomorphic sarcomas of the left atrium, making it difficult to differentiate the pathogenesis and classification of the two entities [14]. In fact, the presence of histologically extensive thrombosis associated with the tumor, as well as its morphological variability and the various immunohistochemical profiles, creates considerable difficulty with the differential diagnosis.

In this study, we collected three cases of IS from two different institutions, plus a local recurrence of one case (case 1) and the tumor xenograft of another (case 3). We performed morphological, immunohistochemical, and molecular studies to find and describe the different morphological and expression patterns in our small series that may be relevant for novel targeted therapies.

## 2. Results

### 2.1. Clinical Findings

All clinical data are summarized in Table 1. The mean age of our cases, two women and one man, was 54 years. Cases 1, 2, and 3 presented multiple local recurrences at 14, 36, and 12 months, respectively, after the first surgery. The surgical margins were not assessed due to specimen fragmentation. Two patients developed lung and cerebral metastasis, and one case spread to the liver. After multiple lines of chemotherapy, survival was 69, 51, and 96 months, respectively, after diagnosis.

### 2.2. Histopathological Findings

Histologically, the tumors presented a diffuse growth pattern with spindle cells and areas of extensive thrombosis (Figure 2). One case presented a more epithelioid morphology with a rich inflammatory stroma presented above all in plasma cells (Figure 2). Areas of myxoid component and focal hyalinization were also found. The tumor xenograft showed cells presenting a spindle configuration with aggressive changes, such as major nuclear pleomorphism, high cellular density, and abundant mitosis (Figure 2). The local recurrence of case 1 displayed similar histological features to the primary neoplasm.

### 2.3. Immunohistochemical Findings

All immunohistochemical results are summarized in Table 2. All cases presented intense immunoexpression for MDM2, CDK4, CD117, c-myc, PDGFRA, and p16, with less intense expression of smooth muscle actin. Aberrant and cytoplasmic expression for GLI-1 and STAT6 was observed in all cases (Figure 3). Podoplanin was positive in two cases, and ERG was positive in one case and in its local recurrence (Figure 3). EMA presented weak and focal expression in one case and in its xenograft. Pan-TRK was positive in one case. Interestingly, p16 expression lost intensity both in local recurrences and in the xenografted tumor (Figure 3). In contrast, PDGFRA, HTERT, and pan-TRK gained expression in both local recurrence and the xenograft (Figure 3). p53 expression was also clearly increased in the xenografted tumor. The Ki67 proliferative index was maintained in the local recurrence but increased in the xenograft. 

### 2.4. Molecular Analysis

The three selected cases showed amplification of *MDM2* by FISH (Figure 4). In patient 1, the NGS panel showed amplifications in the *CDK4* (9.4 copies), *PDGFRA* (8.9 copies), and *KIT* (7.1 copies) genes without detecting SNVs or MSI. The NGS study of the local recurrence from patient 1 was not evaluable due to low sample quality. In patient 2, a *BRAF* V600E mutation (VAF: 4.8% of 5219 readings) and *CDK4* amplification (14.5 copies) were detected. Finally in patient 3, amplifications of the *CDK4* (13.6 copies) and *KRAS* (3.9 copies) genes were detected in the primary tumor. Interestingly, in the tumor xenograft of this last case, mutations in the *GNAQ* exon 4: c.548G>A;p.(Arg183Gln)(R183Q) (VAF: 13.3% of 493 readings), *RAC1* exon 3:c.161G>A;p.(Gly54Asp)(G54D) (VAF:18.2% of 623 readings) and *SF3B1* exon 15: c.2206C>T;p.(Arg736Cys)(R736C) (VAF:13.3% of 1000 readings) genes were identified. 

## 3. Discussion

IS are very rare aggressive mesenchymal tumors with a very poor prognosis despite different lines of chemotherapy. There are not yet any standardized treatments or protocols for this type of cancer; therefore, finding new targeted therapies is crucial to improving outcomes [15]. Very few large series have been published, and the majority of publications in the literature are based on case reports [5]. This tumor also has the ability to differentiate, mimicking other types of sarcomas [6,12,16], confusing the differential diagnosis with other similar spindle cell tumors and thus underestimating the true incidence of this entity [17]. 

Very few series of IS have been studied with a broad panel of immunomarkers [6,18], and recent studies have focused on molecular alterations [14,17,18]. No xenografts have been published. In our wide immunohistochemical panel, we observed intense expression for CD117, c-myc, PDGFRA, and p16 in all cases. To our knowledge p16 expression has not been described before in these kinds of tumors. Interestingly, both the xenograft and the recurrence showed slight loss of p16 intensity. Losses and deletion of *CDKN2A* and *CDKN2B* have been described in some studies [17]. In our few cases, we did not detect deletion of these genes, although loss of immunoexpression was observed. These deletions and the loss of expression of *CDKN2A* and p16 could be factors in disease progression. In contrast, PDGFRA, HTERT, and pan-TRK gained immunoexpression in local recurrence and xenograft samples. HTERT expression has been described as a marker of worse prognosis in solitary fibrous tumors, among others [19,20]. Also interesting is the expression of CD117 and PDGFRA, which creates difficulties for the differential diagnosis with gastrointestinal stromal tumors (GISTs) in some atypical locations. These two markers are correlated with amplifications in *KIT* and *PDGFRA* observed in one of our cases and in other publications [6,14,17,18]. Moreover, these amplifications and overexpression have shown potential as targets for tyrosine kinase inhibitors in IS [13,21]. In fact, Neuville A et al. [6] detected *KIT* and *PDGFRA* amplifications in two of five cases, a similar proportion to our short series.

Few studies in the literature have studied the genomics of this sarcoma [17]. In our small series, we observed co-amplification of *MDM2* and *CDK4* by FISH and NGS. These molecular alterations have been described as essential molecular markers for diagnosis [1]. *MDM2* is a negative regulator of *TP53* located in the 12q12–15 region in the vicinity of the *CDK4*, *HMGA2*, *DDIT3*, and *GLI-1* genes [17,22,23], which are also amplified in IS [6]. In addition, a case report on cardiac intimal sarcoma also found copy gains in *PDGFRA*, *KIT*, *STAT6*, *GLI-1*, *CDK4*, *HMGA2*, and *MDM2* using whole exome sequencing (WES) and array-comparative genomic hybridization (aCGH) [16]. Aberrant cytoplasmic expression was detected for GLI1 and STAT6 in all our cases. This phenomenon could be due to MDM2 region amplification. Agaram NP et al. [23] suggested that *GLI1* amplification may represent an alternative genetic mechanism for GLI1 oncogenic activation akin to *GLI1* fusions, defining the pathogenesis of an emerging group of malignant soft tissue tumors. Moreover, Stein U et al. [24] reported that *GLI* gene amplification is correlated with tumor grade in bone and soft tissue tumors. These authors noted the importance of nuclear staining for GLI1 and STAT6 for differential diagnosis with GLI tumors and solitary fibrous tumors, which can share similar locations and morphology. *MDM2* and *CDK4* amplification is also seen in other malignant tumors, such as dedifferentiated liposarcomas, among others [22], which may create a differential diagnostic challenge, given the morphological, immunohistochemical, and molecular similarities. 

A recent study by Koelsche C et al. [14] described some cases of IS without *MDM2* amplification, which presented mutually exclusive *MDM4* and *CDK6* amplifications. Another recent large series by Yamada Y et al. [18] found IS with *MDM2* amplifications in 55% of cases, associating a myxoid histology with *MDM2* gene amplification and dividing IS into myxoid and non-myxoid types. Thus, it would be important to rule out *MDM4* and *CDK6* amplification in cases with high suspicion of IS by morphology and immunohistochemical profile but without *MDM2* amplification. Nevertheless, controversy exists as to whether these non-amplified tumors are really IS or undifferentiated pleomorphic sarcomas. Koelche C et al. [14] also described amplifications in *PDGFRA*, *MYC*, *MYCN*, and *TERT*. These findings agree with our immunohistochemical profile. C-myc showed intense expression in all our cases, while PDGFRA, HTERT, and pan-TRK gained expression in the local recurrence and the xenografted tumor and so may be indicative of disease progression. Some of these oncogene amplifications in IS may qualify patients for targeted therapies [25,26,27]. Several MDM2 inhibitors are under investigation in clinical trials for the treatment of solid and hematologic tumors [27,28]. However, it remains necessary to determine whether these therapies can lead to treatment response in IS [28,29]. Interestingly, as far as we know, *BRAFs* mutation and *KRAS* amplifications have not been described in IS as actionable genes, allowing the possibility for novel targeted drugs in these patients. Also of interest are the actionable gene mutations acquired and found in tumor progression within nude mice, such as *GNAQ*, *RAC1*, and *SF3B1*. Currently, there are no treatments approved by the FDA for patients with IS mutants for *GNAQ* [30]. The GENIE Project of AACR Consortium has described two mutations in the *RAC1* gene in IS [31]. Finally, mutant cells for *SF3B1* could be sensitive to spliceosome inhibitors [32]. These last findings could create new possibilities for treatment in patients with very poor prognosis.

This study is limited by the small number of cases. Therefore, further studies are needed to confirm the present immunohistochemical and molecular findings regarding this infrequent and challenging entity.

## 4. Materials and Methods

### 4.1. Patients and Samples

Three cases of IS diagnosed between 2010 and 2018 were collected from the Pathology Departments of the Hospital Universitari i Politècnic La Fe of Valencia and the Hospital Clínic Universitari of Valencia. Tissue samples were stained with hematoxylin and eosin (H&E) for histological analysis. Approval for the study was obtained from the Ethics Committee of the Universitat de València Estudi General (UVEG). Clinical data (gender, age, tumor size, tumor and metastasis location, first recurrence, and survival) were also retrieved as shown at Table 1. 

### 4.2. Xenograft

Male nude mice were purchased from IFFA-CREDO^®^ (Lyon, France), kept under specific pathogen-free conditions throughout the experiment, and provided with vinyl isolates plus sterilized food, water, cages, and bedding. The specimen for the xenotransplant was obtained from a surgical specimen of case 3 at the time of diagnosis and placed in culture medium (RPMI 1640) plus antibiotic at 37 °C until transplantation less than 6 h after surgery. Fragments of non-necrotic tumor, about 2 mm in size, were transplanted into the subcutaneous tissue in the backs of two nude mice. The nude mice were sacrificed when the tumor size reached between 15 and 20 mm. The animals were handled in accordance with the regulations of the Ethics Committee of the University of Valencia.

### 4.3. Histopathology

All the available H&E slides were examined by three pathologists (FG, IM, and ALLB), all blinded to the clinical data. In cases of discordant results, a consensus was reached using a multi-head microscope. IS diagnosis was established according to World Health Organization (WHO) criteria. First, we observed the architectural pattern and cellular morphology. We then counted mitoses (number of mitoses per 10 high-power fields [HPFs]). 

### 4.4. Immunohistochemistry

Immunohistochemistry was performed on sections by an indirect peroxidase method as described in Appendix A. Antigen retrieval was performed with heat-induced epitope retrieval (autoclave at 1.5 atmospheres for 3 min in citrate buffer). Bound antibodies were visualized by an avidin-biotin-peroxidase procedure (LSAB Agilent^®^). Appropriate positive and negative controls were used for each antibody. Immunoreactivity was defined as follows: negative (0) with fewer than 5% of tumor cells stained and positive with 5% or more of tumor cells stained. For Ki67 and p53, we recorded the proliferative index and the percentage of immunoexpression, respectively. All sections were independently evaluated by three pathologists (FG, IM, and ALLB), and in cases of disagreement, the score was determined by consensus. 

### 4.5. Fluorescence In Situ Hybridization (FISH)

FISH was performed for the *MDM2* gene evaluation using the dual-color assay MDM2/SE 12 FISH probe ^®^ (Kreatech, Leica Biosystems) mapping in the 12q15 chromosomes region. All formalin-fixed, paraffin-embedded tissue sections (FPPE) were pretreated, digested, and washed using the manual pretreatment Histology Accessory kit according to the manufacturer’s instructions (Dako, Denmark). For *MDM2* gene amplification, a minimum of 100 non-overlapping intact interphase nuclei sections were visualized at 100× magnification with an oil immersion objective using an Axioscope 5 microscope with an Axiocam 305 mono camera and Colibri 5 LED illumination (ZEISS, Carl Zeiss Iberia, S.L, Jena, Germany). Finally, image-processing and FISH image acquisition were performed with the ZEN 3.1 Blue Edition Imaging Software (ZEISS, Carl Zeiss Iberia, S.L). Cases were considered positive for *MDM2* gene amplification when at least 30 of 100 counted tumor cells (30%) showed more than three red signals (number of copies of the *MDM2* gene) and two green signals (Centromere control probe) (3 + R2G).

### 4.6. DNA Isolation

DNA was isolated from five sections of 5 μm-thick FFPE samples using a kitQIAamp^®^ DNA Investigator kit (Qiagen, Hilden, Germany) as indicated by the manufacturer. 

The final concentration was spectrophotometrically measured using Nanodrop^®^ ND-1000 (Eppendorg, Hamburg, Germany).

### 4.7. Next Generation Sequencing (NGS)

The libraries were prepared using the Solid Tumor Solution (STS) gene panel from Sophia Genetics^TM^, following the manufacturer’s recommendations. Briefly, 50 ng of extracted DNA were enzymatically fragmented to a size between 200 and 800 bp. The custom panel interrogates hot spots from 42 genes associated with somatic tumors. The panel includes the following genes; *AKT1*, *ALK*, *BRAF*, *CDK4*, *CDKN2A*, *CTNNB1*, *DDR2*, *DICER1*, *EGFR*, *ERBB2*, *ERBB4*, *FBXW7*, *FGFR1*, *FGFR2*, *FGFR3*, *FOXL2*, *GNA11*, *GNAQ*, *GNAS*, *H3F3A*, *H3F3B*, *HIST1H3B*, *HRAS*, *IDH1*, *IDH2*, *KIT*, *KRAS*, *MAP2K1*, *MET*, *MYOD1*, *NRAS*, *PDGFRA*, *PIK3CA*, *PTPN11*, *RAF1*, *RAF1*, *RET*, *ROS1*, *SF3B1*, *SMAD4*, *TERT*, and *TP53*. The pooled library was sequenced (500 bp/cycles paired-end) on a NextSeq550 (Illumina^TM^). Variant calling and annotation were performed on the Sophia DDM^TM^ platform. Variants were selected based on the following parameters: coverage >600x, allelic frequency (AF) > 10%, and annotation. The variants were filtered based on coverage and functional annotation. The minimum coverage for a variant was established at 600X. Mutations were accepted with a frequency higher than 10%. A germline origin of these mutations was ruled out after germline testing. Visualization was performed with the Integrative genomic viewer (IGV) software. 

## 5. Conclusions

Clinical trial enrollment should be considered for all intimal sarcomas, and further exploration of copy number changes and genetic alterations is needed for this relatively chemotherapy-resistant disease. A global, multicenter, prospective registry/trial would best determine the response of this rare cancer to targeted therapies using a wide molecular panel to study possible actionable genes for each case.

## Figures and Tables

**Figure 1 ijms-24-07535-f001:**
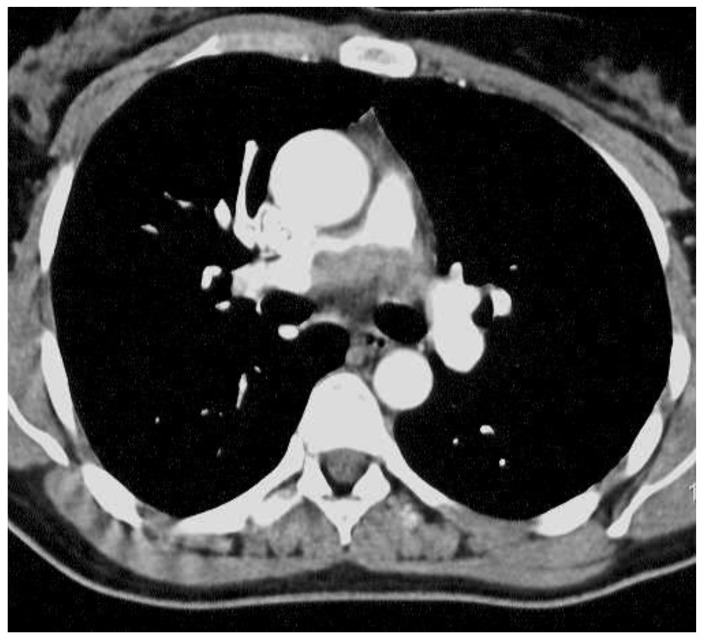
In case 3, CT shows a soft tissue lesion of lobed contours discretely heterogeneously attached to the median wall of the right pulmonary artery, obliterating more than 50% of its lumen.

**Figure 2 ijms-24-07535-f002:**
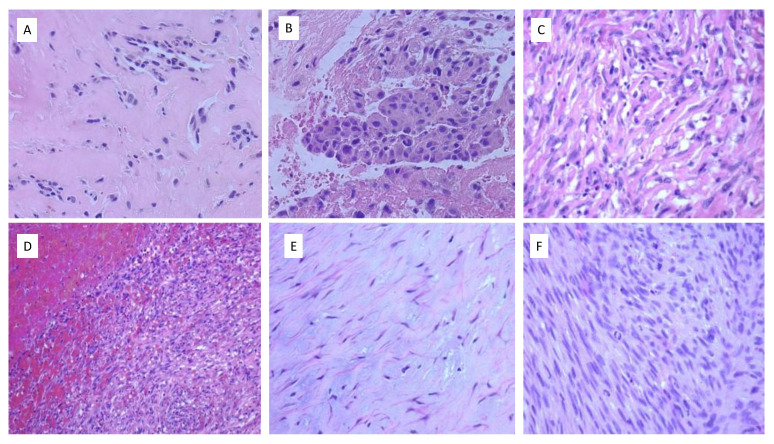
Morphological patterns of IS. (**A**) IS with strong hyalinized areas in the primary tumor of case 1 (H&E, 40×). (**B**) Neoplastic cells with epithelioid features, moderate nuclear pleomorphism, and some mitotic figures (H&E, 40×). (**C**) Spindle neoplastic cells in a fibrous and intense inflammatory stroma in case 2 (H&E, 40×). (**D**) Spindle cell tumor with extensive hemorrhagic areas in case 3 (H&E, 20×). (**E**) Paucicellular tumor areas within a rich myxoid matrix in case 3 (H&E, 40×). (**F**) Dense spindle-cell proliferation with high mitotic activity in the xenografted tumor (H&E, 40×).

**Figure 3 ijms-24-07535-f003:**
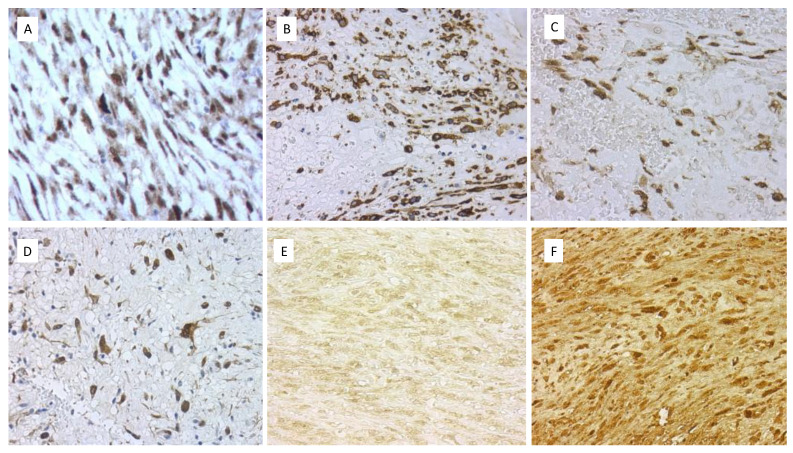
Immunohistochemical markers in different cases of IS. (**A**) High nuclear expression of MDM2 in the primary tumor from case 3 (40×). (**B**) Intense expression of CD117 in case 1 (40×). (**C**) Intense nuclear and cytoplasmic expression of PDGFRA in case 1 (40×). (**D**) Intense nuclear expression of p16 in case 2 (40×). (**E**) Expression of HTERT in the primary tumor from case 3 (40×). (**F**) Higher and intense expression of HTERT in the xenografted tumor (40×).

**Figure 4 ijms-24-07535-f004:**
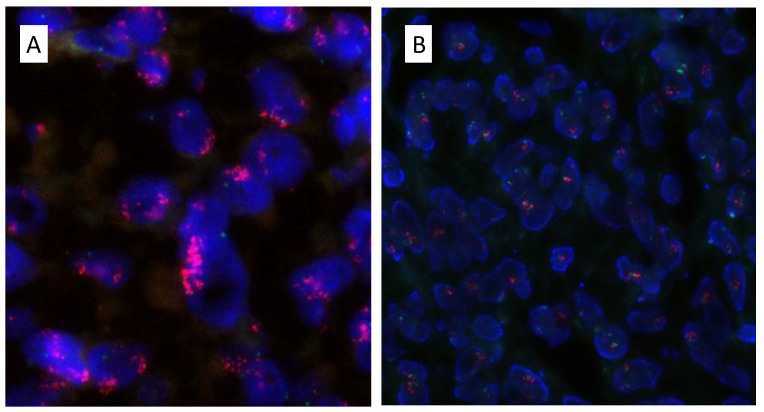
*MDM2* gene amplification by FISH (in red signals). (**A**) *MDM2* gene amplification in case 1 (40×). (**B**) *MDM2* gene amplification in case 3 (20×).

**Table 1 ijms-24-07535-t001:** Demographics and clinical data.

	Case 1	Case 2	Case 3
Age (years)	47	67	47
Gender	M	F	F
Location	Pulmonary artery	Right atrium	Pulmonary artery
First local recurrence (months)	14	36	12
Metastasis location	Lung and liver	Lung and brain	Lung and brain
Follow up (months)	69	51	96
Outcome	DOD	DOD	DOD

M: male; F: female; DOD: died of disease.

**Table 2 ijms-24-07535-t002:** Histopathological features and immunohistochemical profile of intimal sarcomas with xenografted tumor.

	Case 1	Recurrence Case 1	Case 2	Case 3	Xenograft Case 3
Histological pattern	Scant myxoid and pleomorphic tumor with extensive thrombotic areas	Spindle cell and inflammatory (eosinophils and neutrophils) and hyalinized stroma	Epithelioid, pleomorphic, and inflammatory stroma (plasma cells); prominent nucleoli (Reed–Sternberg-like cells)	Spindle and pleomorphic cells; hemorrhagic and thrombotic fields; myxoid and hyalinized	More spindle and pleomorphic features
Mitosis/10HPF	15	10	4	8	6
IHC profile					
MDM2	+	+	+	+	+
CDK4	+	+	+	+	+
Pan-TRK	-	-	-	+	+
H-TERT	+	+	-	+	+
KI67	25%	25%	10%	10%	25%
FLI-1	+	+	+	+	+
CD31	-	-	+	+	+
CD34	-	-	-	-	-
ERG	-	+	-	+	+
CD117	+	+	+	+	+
SMA	-	+	+	+	+
DESMIN	-	-	-	+	+
H-caldesmon	-	-	-	-	-
Factor VIII	-	-	-	+	-
D2.40 (podoplanin)	+	+	-	+	-
C-MYC	+	+	+	+	+
EMA	-	-	-	+/-	+/-
CK(AE1/AE3)	-	-	-	-	-
MyoD1	-	-	-	-	-
p53	2%	1%	5%	5%	25%
p16	+	+	+	+	+
S100	-	-	-	-	-
FOS-B	-	-	-	-	-
EGFR	-	-	-	-	-
PDGFRA	+	+	+	+	+
GLI-1	cytoplasmic	cytoplasmic	cytoplasmic	cytoplasmic	cytoplasmic
STAT-6	cytoplasmic	cytoplasmic	cytoplasmic	cytoplasmic	cytoplasmic

## Data Availability

All data are available.

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
