# Peer review of "Intimal Sarcoma with MDM2/CDK4 Amplification and p16 Overexpression: A Review of Histological Features in Primary Tumor and Xenograft, with Immunophenotype and Molecular Profiling"

_ijms, 2023, doi:10.3390/ijms24087535_

Round 1

Reviewer 1 Report

IS are rare malignant mesenchymal tumors and prognosis is poor.  The authors checked three cases.  They showed that P16 expression loses local recurrence and xenograft.  They concluded two new alterations, a BRAF mutation and KRAS amplification were detected, and new therapeutic options.

Author Response

We would like to thank the reviewer for the evaluation of our manuscript and for their constructive suggestions which we believe have contributed towards improving the article.

With my best regards and thank you very much for evaluating our study.

Reviewer 2 Report

Dear Editor;

This study is a well-written study that can make a contribution to the literature and It evaluate some data that is not available in the  literature. Congratulations to the authors. sincerelly

Author Response

(The authors gave the same response as above.)

Reviewer 3 Report

Comments:

1.     Abstract. Line 23. What does it mean by “P16 lost intensity”? Becomes negative or weaker staining intensity?

2.     It’s odd to put Figure 1 in the Introduction section. Please move it to section 2. Results.

3.     Lines 38-40. This sentence is too lengthy and needs to be revised, maybe as “… intraluminal, leading to local obstruction and …” It’s better to spit into two sentences.

4.     Table 1 is confusing.

5.     Professional English editing is mandatory. For example, Line 96. It’s grammatically incorrect to use “extensive thrombosed areas”. It would be better as “areas of extensive thrombosis”.

6.     The third paragraph of Discussion is too lengthy and is out of focus. This should be restructured into two or three paragraphs.

7.     A paragraph discussing the limitation of this study is advised.

Author Response

We would like to thank the reviewer for the evaluation of our manuscript and for their constructive suggestions which we believe have contributed towards improving the article.

With my best regards and thank you very much for evaluating our study.

Reviewer 3:

  1. Abstract. Line 23. What does it mean by “P16 lost intensity”? Becomes negative or weaker staining intensity?
  • Response: We have specified “weaker” in the sentence.
  1. It’s odd to put Figure 1 in the Introduction section. Please move it to section 2. Results.
  • Response: We have moved Figure 1 to the Results
  1. Lines 38-40. This sentence is too lengthy and needs to be revised, maybe as “… intraluminal, leading to local obstruction and …” It’s better to spit into two sentences.
  • Response: The sentence has been clarified and written in two sentences.
  1. Table 1 is confusing.
  • Response: Table 1 has been clarified.
  1. Professional English editing is mandatory. For example, Line 96. It’s grammatically incorrect to use “extensive thrombosed areas”. It would be better as “areas of extensive thrombosis”.
  • Response: The sentence has been changed in accordance with the reviewer’s suggestion.
  1. The third paragraph of Discussion is too lengthy and is out of focus. This should be restructured into two or three paragraphs.
  • Response: The paragraph has been edited to produce to paragraphs.
  1. A paragraph discussing the limitation of this study is advised.
  • Response: A new paragraph has been added at the end of the Discussion.